# Effect of Aerobic Physical Activity on Health-Related Quality of Life in Middle Aged Women with Osteoarthritis: Korea National Health and Nutrition Examination Survey (2016–2017)

**DOI:** 10.3390/ijerph17020527

**Published:** 2020-01-14

**Authors:** Mikyung Ryu, Sol Lee, Ho Kim, Weon-Chil Baek, Heejin Kimm

**Affiliations:** 1Department of Sport & Leisure Studies, College of Physical Science, Kyonggi University, Suwon 16227, Korea; kyung8545@naver.com; 2Institute on Aging, Ajou University Medical Center, Suwon 16499, Korea; 3Department of bbko Research Center, bbko Big-Data R&D, Seoul 04146, Korea; dlthf1028@naver.com (S.L.); kkhh4@naver.com (H.K.); 4Department of Health Policy and Management, Korea University, Seoul 02841, Korea; 5Department of Epidemiology and Health Promotion, Graduate School of Public Health, Yonsei University, Seoul 03722, Korea; 6Department of Sports and Health Science, College of Human-Centered Convergence, Kyonggi University, Suwon 16227, Korea; bird1028@hanmail.net; 7Department of Epidemiology and Health Promotion, Institute for Health Promotion, Graduate School of Public Health, Yonsei University, Seoul 03722, Korea

**Keywords:** osteoarthritis, health-related quality of life, EQ-5D, aerobic activity, muscle exercise

## Abstract

*Background*: There have been few studies reporting the association between health-related quality of life (HRQoL) and osteoarthritis (OA) in female subjects performing aerobic exercise or not. The aim of this study is to compare HRQoL between OA patients and non-OA subjects in middle aged Korean women (40–59 years) with/without aerobic physical activity, and also to identify an association between EQ-5D instrument based HRQoL and OA controlling for aerobic exercise. *Methods*: This study used the cross-sectional data, the Korea National Health and Nutrition Examination Survey (KNANES) from 2016 to 2017. We only selected those who had completed the evaluations of aerobic physical activity and osteoarthritis diagnosis. In total, 2445 middle aged women were finally selected for this study. The European quality of life-5 dimensions (EQ-5D) was used for the evaluation of HRQoL as a dependent variable. In socio-demographic factors, age, sex, income level, education level, marital status, economic activity, type of insurance and private insurance and behaviour-related factors were included. One-way analysis of variance was conducted to compare the EQ-5D index and multiple linear regression analysis was employed to identify associated factors with the EQ-5D index. *Results*: In total, 2445 middle aged women were included in this study, in which 2209 participants were normal (90.0%) with aerobic physical activity (no: 55.0%, yes: 45.0%) and 246 participants were diagnosed with osteoarthritis (10.0%) with aerobic physical activity (no: 56.1%, yes: 43.9%). In group 2-2 (OA with aerobic), the moderate level of muscle exercise (less than 5 days per week) showed the highest HRQoL (*p* < 0.05) and high level exercise showed the lowest HRQoL (*p* < 0.05). In menopause status, the participants under menopause status showed lower HRQoL (*p* < 0.05) than those not under menopause status in group 2-2 (OA with aerobic) only. *Conclusions*: The HRQoL of OA patients was significantly lower than that of normal participants in middle aged Korean women (40–59 years). Especially, OA patients with maintained aerobic physical activity showed significantly higher HRQoL than those without that under controlling covariates such as age, income level, education level, marital status, economic activity, type of insurance, etc.

## 1. Introduction

Osteoarthritis (OA), causing pain and physical disabilitiy is known as the most common joint disease in the global elderly population due to late-onset degeneration and inflammation of the articular cartilage, which is one of the leading causes of long-term disability with activity limitation and pain, causing low quality of life [1,2,3]. Furthermore, the prevalence of OA dramatically increases with age, as do rates of hospitalization [4]. Moreover, it has been reported that the costs for treatment of OA have come to more than 2000 billion won every year in Korea with increasing social burden [5].

Aerobic physical activities such as walking, jogging, running, and cycling are recommended for health promotion and the prevention of various diseases. Salacinski AJ et al. reported that aerobic exercise showed improved results in gait and pain in individuals with mild-to-moderate knee OA [6]. However, there are also some reports that weight bearing exercise or mechanical loading activity such as treadmill walking or cycling is a possible factor in accelerating the symptoms of OA patients [7,8,9]. Until now, however, there have been few studies determining an optimal aerobic exercise protocol which can enhance health conditions without causing any kind of joint problems or OA symptoms. For this reason, it is not recommended for a patient to engage in weight bearing aerobic activities.

Health-related quality of life (HRQoL) shows physical, cognitive, emotional, and social aspects measured by questionnaires, and evaluates the effect of a disease, disability, or disorder on individual’s life quality [10]. Until now, a lot of HRQoL questionnaires have been developed and are used to evaluate life quality in patients with various diseases or disabilities [11,12,13,14], in which there are two types of instruments—specific instruments and generic instruments. As a generic measure, European quality of life-5 dimensions (EQ-5D) was developed by the EuroQol Group to evaluate [15].

Until now, there have been few studies reporting the association between HRQoL and OA in female subjects performing aerobic exercise or not. The aim of this study is to compare HRQoL between OA patients and non-OA subjects in middle aged Korean women (40–59 years) with/without aerobic physical activity, and also to identify an association between EQ-5D instrument based HRQoL and OA controlling for aerobic exercise. 

## 2. Methods

### 2.1. Study Population and Data Collection

This study used cross-sectional data from the Korea National Health and Nutrition Examination Survey (KNANES) from 2016 to 2017, which was acquired by the Korea Centers for Disease Control and Prevention (KCDC); the details of this study design and data resource profiles were described by Kweon et al. in 2014 [16]. Detailed descriptions of the methods followed the Guidelines for Use of KNHANES Raw Data and the Final Report of KNHANES sampling frame [16]. Multistage stratified cluster sampling was performed for the household unit selection, and the Institutional Review Board of the Korea Centers for Disease Control and Prevention approved the protocols of this study (11702), obtaining written informed consent forms from all participants. KNHANES data (2016–2017) was used in this study, which included the survey results from a health-related interview, medical examination, and various nutritional assessments. All analytical procedures followed the guidelines for the use of data from KCDC. 

We included middle-aged (range: 40–59 years) female participants who completed the following assessments: aerobic physical activity (yes or no) and osteoarthritis diagnosis (yes or no). Participants who were previously diagnosed with OA and/or treated by a physician were included in the OA group; participants who were not diagnosed with OA by physicians via health survey questionnaires in KNHANES were included in the non-OA group (control group). The participants who did not provide an answer, responded to the questionnaire as non-applicable, and were not middle-aged (40–59 years) were excluded from the study (Figure 1). In total, 2445 middle aged women were finally selected in this study.

The European quality of life-5 dimensions (EQ-5D) was used for the evaluation of HRQoL in this study as a dependent variable. EQ-5D was designed to measure and assess health. It includes the five dimensions of health, such as mobility, self-care, routine activities, pain-discomfort, and anxiety/depression. Each dimension has 3 response categories corresponding to no problems, some problems, and extreme problems. Nam et al. estimated the weight for a quality of life survey using EQ-5D in 2007 [17]. A score of 1, the highest in the EQ-5D index, is considered to be the best health condition, but lower scores indicate a lower HRQoL. The EQ-5D has been widely tested and used in both the general population and patient samples and has been translated into more than 130 different languages. EQ-5D has already been validated as an HRQoL assessment tool in OA patients in 2010 [18]. In OA patients, EQ-5D has already been validated as an HRQoL assessment tool in 2010. 

Regarding socio-demographic factors, age (range: 40~59), sex (female only), income (level 1, level 2, level 3, or level 4) education (elementary, middle, high school, or above university), marital status (single, married, or divorced), economic activity (yes or no), type of insurance (local government, company, or national health insurance), and private insurance (yes or no) were included. Health behavior-related factors such as self-related health status (good, fair, or bad), sleeping time (<7, 7+, or 8+ hours per day), drinking (never experience, no drink for 1 year, less than 4 times per month, or more 4 times per month), smoking (non-smoker, past-smoker, or present-smoker), and muscle exercise (no-muscle exercise, less than 5 days per week, or more than 5 days per week) were included as an independent variables. Menopause status (menopause or no-menopause) and obesity (low, normal, or obesity) were also included as independent variables for analysis.

### 2.2. Statistics

For describing the general characteristics of the participants, the frequency in each group, OA with/without aerobic physical activity and non-OA with/without aerobic physical activity, was compared by the chi-square test. One-way analysis of variance (one-way ANOVA) was conducted to compare the EQ-5D index, and multiple linear regression analysis was employed to identify factors associated with the EQ-5D index. The statistical significance was set at 5% in this study.

## 3. Results

In total, 2445 middle aged women were included in this study (Table 1), in which 2209 participants were normal (90.0%) with aerobic physical activity (no = 55.0% and yes = 45.0%) and 246 participants were diagnosed with osteoarthritis (10.0%) with aerobic physical activity (no = 56.1% and yes = 43.9%). All variables in the non-OA group showed a statistically significant difference between the aerobic and non-aerobic physical activity groups (*p* < 0.05). In the OA group, level of income, education, drinking, smoking, sleep time, and muscle exercise showed statistically significant differences between the aerobic and non-aerobic physical activity groups (*p* < 0.05).

In group 1-1 (non-OA with non-aerobic activity), there were significant differences in age, income level, education level, marital status, economic activity, type of insurance, self-related health status, menopause status, and obesity (*p* < 0.05), and most variables in group 1-2 (non-OA with aerobic) showed significant differences (*p* < 0.05) (Table 2). In group 2-1 (OA with non-aerobic), there were significant differences in income level, marital status, economic activity, type of insurance, self-related health status, and muscle exercise (*p* < 0.05), and only type of insurance and self-related health status were significantly different in group 2-2 (OA with aerobic) (*p* < 0.05) (Table 2).

Table 3 showed results of multiple linear regression for the EQ-5D. In group 1-1 and 1-2, normal participants in the lowest education level (elementary school) showed lower HRQoL (*p* < 0.05) than did those in the highest education level (university) and the participants with no economic activity reported lower HRQoL (*p* < 0.05). In self-related health status, the participants with responding “good” showed the highest HRQoL (*p* < 0.05) in both group 1-1 and 1-2. In type of insurance, the participants with company insurance showed the highest HRQoL in both group 1-1 and 1-2 (*p* < 0.05), and group 1-2 showed the lowest HRQoL in no drinking for 1 year (*p* < 0.05). In sleeping time, more than 8 h sleep showed the highest HRQoL (*p* < 0.05) than less than 7 h sleep in group 1-1 only. In group 2, the lowest income level showed the lowest HRQoL (*p* < 0.05) in group 2-1 only and then in type of insurance, the participants with local government insurance showed the highest HRQoL (*p* < 0.05) in group 2-1 only. In self-related health status, the participants with responding “good” showed the highest HRQoL (*p* < 0.05) in both group 1-1 and 1-2 as group 1 showed. The participants sleeping more than 8 h per day showed the significant highest HRQoL in group 2-2 only and in muscle exercise, the moderate level (less than 5 days per week) showed the highest HRQoL (*p* < 0.05) and high level exercise showed the lowest HRQoL (*p* < 0.05). In menopause status, the participants under menopause status showed lower HRQoL (*p* < 0.05) than those not under menopause status in group 2-2 only.

## 4. Discussion

This research was performed to compare HRQoL between OA patients and non-OA subjects in middle aged Korean women (aged from 40 to 59 years old) controlling for aerobic physical activity, and also to identify an association between EQ-5D instrument based HRQoL and OA, considering aerobic exercise, by using KNHANES data (2016–2017 year).

OA is known as a relatively common musculoskeletal disorder with a high prevalence, increasing with age [19], and there are also gender specific differences which mean that OA is more prevalent in female individuals than male [20,21,22,23]. In previous researches, pain, physical disabilities, decreased mobility (function), and mental health status due to OA have been proven to decrease HRQol of OA patients [24,25,26]. Especially, Yang et al. reported that HRQoL, evaluated with the EQ-5D index, could be affected more negatively by OA in Korean elderly participants in 2017 [27]. In this research, it was demonstrated that OA could affect HRQoL using EQ-5D in middle aged female participants controlling for socio-economic factors and health behaviour-related factors.

Until now, it has been demonstrated that aerobic physical exercise such as walking, jogging, running, and cycling are recommended to improve health conditions and prevent various diseases. Moreover, high or low-intensity aerobic physical exercise was proven to have both short and long-term advantages in improving functional status, gait, pain, and aerobic capacity in patients with knee OA [28]. On the other hand, there have been few articles that have been suggested an optimal aerobic physical activity protocol which can enhance health-related conditions while decreasing the possibility of any kind of joint problems or OA symptoms. However, practicing aerobic physical activity was proven to be effective in improving HRQoL in this study.

In 2015, Fransen et al. reported that land-based therapeutic exercise is beneficial for people with knee OA in terms of reducing joint pain or improving physical function and quality of life.

Several studies have shown that Vitamin D deficiency is associated with reduced articular cartilage thickness, the risk of cartilage degeneration, and the onset of OA. In addition, Szychlinska MA et al. (2019) reported vitamin D supplements as a non-pharmacological treatment in early OA patients with a non-established diagnosis [29].

Moreover, high and low intensity aerobic exercises are equally possible to affect the improvement of physical function, walking, and pain [28,30,31,32]. As several previous studies have reported, there is a consensus that the key process to improve HRQoL of the patients with OA is to release pain and to cover physical function [33,34,35,36,37,38], for which various types of exercises would be recommendable. Especially, several systematic reviews showed that exercises including muscle strengthening or aerobic exercises could be effective for patients with OA [28,30,32]. In our research, keeping aerobic physical activity was identified to affect HRQoL in the patient with OA, which means that a low HRQoL could be raised by aerobic physical activity.

This study has several limitations in interpreting the results. First of all, we could not demonstrate any causal relationship between change in HRQoL and aerobic physical activity in OA patients aged from 40 to 59 years old because the data are not appropriate for the study of causality. Second, it was not possible to adjust whether OA patients were undergoing treatment by any intervention or medication, because the data used in this study did not include those survey categories. Finally, there was no consideration of types of aerobic physical exercise. In the future, it is necessary to overcome those limitations. Further prospective studies with a long follow-up period can be conducted to evaluate the causal effect of aerobic physical activity on HRQoL in middle aged Korean women with OA.

## 5. Conclusions

This study demonstrated that HRQoL of OA patients was significantly lower than that of normal participants in middle aged Korean women (40–59 y). Especially, OA patients who maintained aerobic physical activity showed significantly higher HRQoL than those who did not, controlling for covariates such as age, income level, education level, marital status, economic activity, type of insurance, etc.

## Figures and Tables

**Figure 1 ijerph-17-00527-f001:**
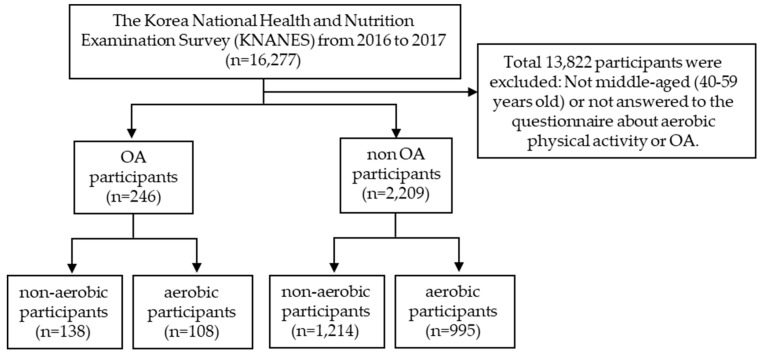
Flowchart describing the selection criteria for the participants.

**Table 1 ijerph-17-00527-t001:** Characteristics of subjects included study.

Item		non-OA ^a^ (*n* = 2209)					OA ^a^ (*n* = 246)				
		Non-Aerobic (*n* = 1214)		Aerobic (*n* = 995)		*p*-Value	Non-Aerobic (*n* = 138)		Aerobic (*n* = 108)		*p*-Value
		*n*	%	*n*	%		*n*	%	*n*	%	
Age (years)	>40	631	52.0	520	52.3	0.034 *	26	18.8	18	16.7	0.121
	<59	583	48.0	475	47.7		112	81.2	90	83.3	
Income level	level 1	307	25.3	214	21.5	<0.001 **	46	33.3	35	32.4	0.046 *
	level 2	322	26.6	237	23.8		35	25.4	25	23.1	
	level 3	287	23.7	260	26.1		30	21.7	30	27.8	
	level 4	296	24.4	284	28.5		27	19.6	18	16.7	
Education level	elementary	110	9.1	58	5.8	<0.001 **	42	30.4	18	16.7	0.006 **
	middle	130	10.7	89	9.0		26	18.8	22	20.4	
	high	526	43.4	425	42.8		46	33.3	48	44.4	
	university	445	36.7	422	42.5		24	17.4	20	18.5	
Marital status	single	37	3.1	35	3.5	0.004 **	1	0.7	1	0.9	0.097
	marriaged	1022	84.3	864	86.9		109	79.0	89	82.4	
	divorced	154	12.7	95	9.6		28	20.3	18	16.7	
Economic activity	no	417	34.4	414	41.6	<0.001 **	52	37.7	39	36.1	0.103
	yes	796	65.6	581	58.4		86	62.3	69	63.9	
Type of insurance	local	408	33.7	253	25.5	<0.001 **	48	34.8	34	31.5	0.087
	company	764	63.2	719	72.5		78	56.5	70	64.8	
	gov. cover	37	3.1	20	2.0		12	8.7	4	3.7	
Private insurance	no	97	8.0	66	6.6	0.031 *	20	14.5	10	9.3	0.074
	yes	1115	92.0	928	93.4		118	85.5	98	90.7	
Self-health status	good	301	24.8	323	32.5	<0.001 **	26	18.8	16	14.8	0.074
	fair	704	58.0	531	53.4		68	49.3	60	55.6	
	bad	209	17.2	141	14.2		44	31.9	32	29.6	
Drinking	non-drinker	139	11.5	91	9.2	0.003 **	20	14.5	19	17.6	0.041 *
	1 year non-drinker	215	17.8	149	15.0		33	23.9	20	18.5	
	<4/m ^b^	705	58.3	634	63.8		64	46.4	62	57.4	
	>4/m ^b^	151	12.5	120	12.1		21	15.2	7	6.5	
Smoking	non-smoker	1081	89.4	918	92.4	0.001 **	120	87.0	101	93.5	0.035 *
	past	57	4.7	43	4.3		9	6.5	3	2.8	
	present	71	5.9	32	3.2		9	6.5	4	3.7	
Sleep hours (per day)	<7	507	41.8	446	44.8	0.003 **	59	42.8	55	50.9	0.032 *
	7	386	31.8	325	32.7		44	31.9	30	27.8	
	≥8	320	26.4	224	22.5		35	25.4	23	21.3	
Muscle exercise (days/week)	no	1057	87.1	736	74.0	<0.001 **	116	84.1	82	75.9	0.031 *
	<5	117	9.6	192	19.3		13	9.4	16	14.8	
	>5	39	3.2	67	6.7		9	6.5	10	9.3	
Menopause	menopause	531	44.0	431	43.4	0.033 *	106	76.8	84	77.8	0.120
	no-menopause	677	56.0	563	56.6		32	23.2	24	22.2	
Obesity	low	42	3.5	33	3.3	0.030 *	3	2.2			0.084
	normal	818	67.4	684	68.7		78	56.5	70	64.8	
	obesity	354	29.2	278	27.9		57	41.3	38	35.2	

^a^ OA = osteoarthritis, ^b^ 1/m = 1 glass per month, * *p* < 0.05, ** *p* < 0.01.

**Table 2 ijerph-17-00527-t002:** Relationship between aerobic activity and subject characteristics in osteoarthritis and non-osteoarthritis.

Item		Group 1 Non-OA ^a^ (*n* = 2209)						Group 2 OA ^a^ (*n* = 246)					
		Group 1-1 Non-Aerobic (*n* = 1214)		*p*-Value	Group1-2 Aerobic (*n* = 995)		*p*-Value	Group2-1 Non-Aerobic (*n* = 138)		*p*-Value	Group2-2 Aerobic (*n* = 108)		*p*-Value
		Mean	SD		Mean	SD		Mean	SD		Mean	SD	
Age (years)	>40	0.97	0.06	0.006 **	0.98	0.06	0.011 *	0.89	0.20	0.869	0.92	0.11	0.749
	<59	0.96	0.07		0.97	0.07		0.90	0.15		0.91	0.11	
Income level	level 1	0.95	0.09	<0.001 **	0.96	0.10	<0.001 **	0.82	0.23	<0.001 **	0.89	0.12	0.166
	level 2	0.97	0.06		0.97	0.06		0.95	0.09		0.91	0.11	
	level 3	0.97	0.06		0.98	0.05		0.91	0.10		0.94	0.08	
	level 4	0.98	0.05		0.98	0.06		0.95	0.06		0.94	0.08	
Education level	elementary	0.95	0.09	<0.001 **	0.92	0.12	<0.001 **	0.85	0.23	0.076	0.88	0.14	0.139
	middle	0.95	0.08		0.95	0.11		0.91	0.13		0.90	0.10	
	high	0.97	0.06		0.98	0.05		0.93	0.10		0.94	0.09	
	university	0.98	0.05		0.98	0.05		0.93	0.13		0.91	0.11	
Marital status	single	0.97	0.06	0.001 **	0.96	0.09	0.002 **	1.00	.	<0.001 **	1.00	.	0.442
	marriaged	0.97	0.06		0.97	0.06		0.93	0.09		0.92	0.10	
	divorced	0.95	0.09		0.95	0.11		0.79	0.29		0.89	0.13	
Economic activity	no	0.96	0.08	0.001 **	0.96	0.08	0.003 **	0.83	0.22	<0.001 **	0.90	0.12	0.306
	yes	0.97	0.06		0.98	0.06		0.94	0.08		0.92	0.09	
Type of insurance	local	0.97	0.07	<0.001 **	0.97	0.07	<0.001 **	0.93	0.09	<0.001 **	0.88	0.12	0.014 *
	company	0.97	0.06		0.98	0.06		0.92	0.15		0.94	0.09	
	gov. cover	0.90	0.12		0.86	0.20		0.64	0.23		0.85	0.07	
Private insurance	no	0.96	0.10	0.086	0.95	0.11	0.003 **	0.87	0.15	0.385	0.91	0.10	0.841
	yes	0.97	0.06		0.97	0.06		0.90	0.16		0.92	0.11	
Self-health status	good	0.99	0.03	<0.001 **	0.99	0.03	<0.001 **	0.96	0.07	<0.001 **	0.96	0.06	<0.001 **
	fair	0.97	0.05		0.98	0.06		0.95	0.07		0.94	0.07	
	bad	0.92	0.11		0.91	0.12		0.78	0.23		0.85	0.14	
Drinking	non-drinker	0.97	0.06	0.319	0.97	0.06	0.022 *	0.89	0.22	0.603	0.91	0.12	0.953
	1 year non-drinker	0.96	0.08		0.96	0.08		0.89	0.09		0.92	0.12	
	<4/m ^b^	0.97	0.06		0.97	0.07		0.90	0.19		0.92	0.09	
	>4/m ^b^	0.96	0.07		0.98	0.04		0.94	0.07		0.91	0.15	
Smoking	non-smoker	0.97	0.06	0.154	0.97	0.07	0.406	0.90	0.16	0.409	0.92	0.10	0.236
	past	0.96	0.07		0.97	0.06		0.90	0.11		0.82	0.23	
	present	0.96	0.08		0.96	0.09		0.83	0.19		0.88	0.19	
Sleep hours (per day)	<7	0.97	0.07	0.242	0.97	0.07	0.340	0.88	0.21	0.591	0.90	0.12	0.144
	7	0.97	0.06		0.98	0.07		0.91	0.11		0.94	0.09	
	≥8	0.97	0.06		0.97	0.07		0.92	0.12		0.93	0.08	
Muscle exercise (days/week)	no	0.97	0.07	0.411	0.97	0.07	0.300	0.90	0.13	0.047 *	0.91	0.11	0.902
	<5	0.97	0.06		0.98	0.06		0.94	0.10		0.93	0.10	
	>5	0.98	0.04		0.98	0.04		0.78	0.39		0.92	0.10	
Menopause	menopause	0.96	0.07	0.001 **	0.96	0.08	0.001 **	0.90	0.16	0.750	0.91	0.11	0.082
	no-menopause	0.97	0.06		0.98	0.06		0.91	0.18		0.95	0.07	
Obesity	low	0.96	0.06	<0.001 **	0.97	0.08	0.041 *	0.93	0.07	0.946	0.92	0.10	0.551
	normal	0.97	0.06		0.98	0.06		0.90	0.17		0.91	0.11	
	obesity	0.96	0.08		0.96	0.08		0.90	0.16		0.92	0.11	

^a^ OA = osteoarthritis, ^b^ 1/m = 1 glass per month, * *p* < 0.05, ** *p* < 0.01.

**Table 3 ijerph-17-00527-t003:** Multiple linear regression analysis between osteoarthritis and aerobic activity.

Item		non-OA ^a^ (*n* = 2209)						OA ^a^ (*n* = 246)					
		Non-Aerobic (*n* = 1214)			Aerobic (*n* = 995)			Non-Aerobic (*n* = 138)			Aerobic (*n* = 108)		
Age (years)	>40	0.003	0.005	0.543	−0.001	0.006	0.797	−0.032	0.036	0.376	−0.043	0.032	0.182
	<59	-			-			-			-		
Income level	level 1	−0.002	0.005	0.743	−0.003	0.006	0.667	−0.067	0.033	0.041 *	−0.039	0.032	0.216
	level 2	0.002	0.005	0.641	−0.001	0.005	0.854	−0.020	0.031	0.528	−0.054	0.030	0.067
	level 3	0.002	0.005	0.661	0.002	0.005	0.760	−0.045	0.034	0.185	−0.010	0.029	0.723
	level 4	-			-			-			-		
Education level	elementary	−0.016	0.007	0.020 *	−0.025	0.010	0.008 **	−0.019	0.034	0.585	0.007	0.036	0.847
	middle	−0.008	0.006	0.223	−0.016	0.008	0.045 *	0.024	0.035	0.493	0.008	0.031	0.792
	high	0.001	0.004	0.731	0.002	0.004	0.602	0.033	0.032	0.297	0.016	0.026	0.543
	university	-			-			-			-		
Marital status	single	0.010	0.011	0.389	−0.001	0.012	0.938	0.039	0.150	0.792	−0.003	0.097	0.975
	marriaged	0.007	0.006	0.183	0.005	0.007	0.467	0.025	0.030	0.401	0.013	0.026	0.618
	divorced	-			-			-			-		
Economic activity	no	−0.008	0.004	0.030 *	−0.009	0.004	0.021 *	−0.041	0.024	0.084	−0.026	0.020	0.212
	yes	-			0 ^a^			0 ^a^			0 ^a^		
Type of insurance	local	0.049	0.011	<0.001 **	0.086	0.015	<0.001 **	0.123	0.047	0.008 **	−0.016	0.048	0.737
	company	0.054	0.011	<0.001 **	0.090	0.015	<0.001 **	0.101	0.046	0.027 *	0.053	0.047	0.263
	gov. cover	-			-			-			-		
Private insurance	no	0.001	0.007	0.895	−0.004	0.008	0.643	0.030	0.030	0.312	−0.009	0.034	0.793
	yes	-			-			-			-		
Self-health status	good	0.069	0.006	<0.001 **	0.062	0.007	<0.001 **	0.120	0.032	<0.001 **	0.122	0.031	<0.001 **
	fair	0.055	0.005	<0.001 **	0.052	0.006	<0.001 **	0.135	0.024	<0.001 **	0.093	0.021	<0.001 **
	bad	-			-			-			-		
Drinking	non-drinker	0.013	0.007	0.062	−0.008	0.009	0.350	0.002	0.040	0.958	0.014	0.043	0.747
	1 year non-drinker	0.009	0.006	0.157	−0.015	0.008	0.049	−0.001	0.035	0.969	0.025	0.040	0.537
	<4/m ^b^	0.007	0.005	0.198	−0.011	0.006	0.078	0.006	0.031	0.841	0.012	0.039	0.766
	>4/m ^b^	-			-			-			-		
Smoking	non-smoker	−0.006	0.008	0.400	0.003	0.011	0.765	−0.018	0.041	0.670	−0.021	0.051	0.678
	past	−0.008	0.011	0.453	0.001	0.014	0.933	0.015	0.055	0.781	−0.056	0.073	0.444
	present	-			-			-			-		
Sleep hours (per day)	<7	−0.010	0.004	0.020 *	0.001	0.005	0.818	−0.013	0.026	0.624	−0.048	0.022	0.030 *
	7	−0.004	0.005	0.436	0.004	0.005	0.417	−0.014	0.027	0.604	−0.030	0.027	0.270
	≥8	-			-			-			-		
Muscle exercise (days/week)	no	−0.008	0.010	0.387	−0.010	0.008	0.193	0.141	0.042	0.001 **	0.034	0.030	0.256
	<5	−0.009	0.011	0.413	−0.011	0.009	0.211	0.173	0.053	0.001 **	0.029	0.035	0.408
	>5	-			-			-			-		
Menopause	menopause	−0.003	0.005	0.482	−0.006	0.006	0.298	0.006	0.031	0.858	−0.079	0.028	0.005 **
	no-menopause	-			-			-			-		
Obesity	low	−0.005	0.010	0.623	0.001	0.011	0.945	0.015	0.088	0.867			
	normal	0.006	0.004	0.127	0.003	0.004	0.557	−0.003	0.022	0.884	−0.021	0.018	0.259
	obesity	-			-			-			-		

^a^ OA = osteoarthritis, ^b^ 1/m = 1 glass per month, * *p* < 0.05, ** *p* < 0.01.

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
