# Peer review of "Effect of Aerobic Physical Activity on Health-Related Quality of Life in Middle Aged Women with Osteoarthritis: Korea National Health and Nutrition Examination Survey (2016–2017)"

_ijerph, 2020, doi:10.3390/ijerph17020527_

Round 1

Reviewer 1 Report

Manuscript titled “Effect of aerobic physical activity on health-related quality of life in middle aged women with osteoarthritis: Korea National Health and Nutrition Examination Survey (2016-2017).” deals an important issue of osteoarthritis. The aim of this study is to compare HRQoL between OA patient and non-OA subject in middle aged Korean women (40y - 59y) with/without aerobic physical activity and also to identify an association between EQ-5D instrument based HRQoL and OA with controlling aerobic exercise.

The work is great and complete. Moreover, there are some minor concerns that need to be addressed before recommending publication.

It would be very interesting if the authors added in the text, to help better readers understanding, a sentence regarding the importance of physical activity and nutrition (Vitamin D, Mediterranean Diet) as a non-pharmacologic treatment in OA.  Please quote these interesting and recent papers or other similar:

Assessment of Vitamin D Supplementation on Articular Cartilage Morphology in a Young Healthy Sedentary Rat Model. Nutrients. 2019 Jun 3;11(6).

In the conclusion section please specify better the clinical relevance of your work.

Author Response

Response to reviewer comments

January 9, 2020

Sara Xia

Dear Editor,

Thank you for your kind letter dated January 3, 2020 regarding our manuscript entitled “Effect of aerobic physical activity on health-related quality of life in middle aged women with osteoarthritis: Korea National Health and Nutrition Examination Survey (2016-2017).” We have carefully read the suggestions of the reviewers and wish to resubmit the manuscript after revisions. We have revised the manuscript as per the inputs and suggestions of the reviewers (changes in the revised manuscript are indicated using the “Track Changes” feature of MS Word). We would like to express our sincere gratitude and appreciation for the valuable comments that have helped us to significantly improve the manuscript. We have provided point-by-point responses to the reviewer’s comments. We hope that the revised manuscript will be suitable for publication in International Journal of Environmental Research and Public Health.

Sincerely yours,

Heejin Kimm, M.D., Ph.D.

Department of Epidemiology and Health Promotion,

Graduate School of Public Health, Yonsei University,

Seoul, South Korea, 120-752

---------------------------------------------------------------------------------------

Response to the comments of the reviewers

Reviewer # 1

Manuscript titled “Effect of aerobic physical activity on health-related quality of life in middle aged women with osteoarthritis: Korea National Health and Nutrition Examination Survey (2016-2017).” deals an important issue of osteoarthritis. The aim of this study is to compare HRQoL between OA patient and non-OA subject in middle aged Korean women (40y - 59y) with/without aerobic physical activity and also to identify an association between EQ-5D instrument based HRQoL and OA with controlling aerobic exercise.

The work is great and complete. Moreover, there are some minor concerns that need to be addressed before recommending publication.

It would be very interesting if the authors added in the text, to help better readers understanding, a sentence regarding the importance of physical activity and nutrition (Vitamin D, Mediterranean Diet) as a non-pharmacologic treatment in OA. Please quote these interesting and recent papers or other similar:

 Assessment of Vitamin D Supplementation on Articular Cartilage Morphology in a Young Healthy Sedentary Rat Model. Nutrients. 2019 Jun 3;11(6).

Response: We have included the abovementioned paper as a reference [Nutrients. 2019 Jun 3;11(6).]

Several studies have shown that Vitamin D deficiency is associated with reduced articular cartilage thickness, the risk of cartilage degeneration, and the onset of OA. In addition, Szychlinska MA et al. (2019) reported vitamin D supplements as a non-pharmacological treatment in early OA patients with a non-established diagnosis.

In the conclusion section please specify better the clinical relevance of your work.

Response: We have added some texts in the conclusion section.

This study demonstrated that HRQoL of OA patients was significantly lower than that of normal participants in middle aged Korean women (40y - 59y). Especially, OA patients with keeping aerobic physical activity showed significantly higher HRQoL than those without that under controlling covariates such as age, income level, education level, marital status, economic activity, type of insurance, etc. Further prospective studies with a long follow-up period can be conducted to evaluate the causal effect of aerobic physical activity on HRQoL in middle aged Korean women with OA.

Reviewer 2 Report

COMMENTS TO AUTHORS:

It is a very important study in this field. This paper to compare HRQoL between OA patient and non-OA subject in middle aged Korean women (40y - 59y) with/without aerobic physical activity and also to identify an association between EQ-5D instrument based HRQoL and OA with controlling aerobic exercise. They also demonstrated that HRQoL of OA patients was significantly lower than that of normal 196 participants in middle aged Korean women (40y - 59y). I do have some comments as listed below in the order noted.

Comment 1:

The quality of the data set is very important, especially in a Korea National Health and Nutrition Examination Survey (KNANES)-based people. For this reason, please clarify the included criteria and excluded criteria of sample collection in the Study Population and Data Collection section and please provide a flowchart at the end of the subsection.

Comment2:

Please clarify and define the European quality of life-5 dimensions (EQ-5D) in the subsection of Study Population and Data Collection.

Author Response

Response to reviewer comments

January 9, 2020

Sara Xia

Dear Editor,

Thank you for your kind letter dated January 3, 2020 regarding our manuscript entitled “Effect of aerobic physical activity on health-related quality of life in middle aged women with osteoarthritis: Korea National Health and Nutrition Examination Survey (2016-2017).” We have carefully read the suggestions of the reviewers and wish to resubmit the manuscript after revisions. We have revised the manuscript as per the inputs and suggestions of the reviewers (changes in the revised manuscript are indicated using the “Track Changes” feature of MS Word). We would like to express our sincere gratitude and appreciation for the valuable comments that have helped us to significantly improve the manuscript. We have provided point-by-point responses to the reviewer’s comments. We hope that the revised manuscript will be suitable for publication in International Journal of Environmental Research and Public Health.

Sincerely yours,

Heejin Kimm, M.D., Ph.D.

Department of Epidemiology and Health Promotion,

Graduate School of Public Health, Yonsei University,

Seoul, South Korea, 120-752

-------------------------------------------------------------------------------------

Response to the comments of the reviewers

Reviewer # 2

COMMENTS TO AUTHORS:

It is a very important study in this field. This paper to compare HRQoL between OA patient and non-OA subject in middle aged Korean women (40y - 59y) with/without aerobic physical activity and also to identify an association between EQ-5D instrument based HRQoL and OA with controlling aerobic exercise. They also demonstrated that HRQoL of OA patients was significantly lower than that of normal 196 participants in middle aged Korean women (40y - 59y). I do have some comments as listed below in the order noted.

Comment 1:

The quality of the data set is very important, especially in a Korea National Health and Nutrition Examination Survey (KNANES)-based people. For this reason, please clarify the included criteria and excluded criteria of sample collection in the Study Population and Data Collection section and please provide a flowchart at the end of the subsection.

Response: We have clarified the inclusion and exclusion criteria in the Study Population and Data Collection sections and have inserted a selection criteria flowchart to aid reader comprehension.

We included middle-aged (range: 40-59 years) female participants who completed the following assessments: aerobic physical activity (yes or no) and osteoarthritis diagnosis (yes or no). Participants who were previously diagnosed with OA and/or treated by a physician were included in the OA group; participants who were not diagnosed with OA by physicians via health survey questionnaires in KNHANES were included in the non-OA group (control group). The participants who did not provide an answer, responded to the questionnaire as non-applicable, and were not middle-aged (40-59 years) were excluded from the study (Fig. 1).

Figure 1. Flowchart describing the selection criteria for the participants

<Please see the attachment>

Comment2:

Please clarify and define the European quality of life-5 dimensions (EQ-5D) in the subsection of Study Population and Data Collection.

Response: We have clarified and defined the EQ-5D in the Study Population and Data Collection.

The European quality of life-5 dimensions (EQ-5D) was used for the evaluation of HRQoL in this study as a dependent variable. EQ-5D was designed to measure and assess health. It includes the five dimensions of health, such as mobility, self-care, routine activities, pain-discomfort, and anxiety-depression. Each dimension has 3 response categories corresponding to no problems, some problems, and extreme problems. Nam et al estimated weight for quality of life survey using EQ-5D in 200717. Score 1, the highest EQ-5D index, is considered to be the best health condition, but lower scores indicate a lower HRQoL. The EQ-5D has been widely tested and used in both general population and patient samples and has been translated into more than 130 different languages. EQ-5D has already been validated as an HRQoL assessment tool in OA patients in 2010.